# Emission Reduction via Fossil Fuel Subsidy Removal and Carbon Pricing, Creating Synergies with Revenue Recycling

**Andrea Marcello Bassi [1,2,]*[ID], Georg Pallaske [1], Richard Bridle [2] and Kavya Bajaj [2]**

1    KnowlEdge Srl, 21057 Olgiate Olona, Italy
2    International Institute for Sustainable Development (IISD), Winnipeg, MB R3B 0T4, Canada
*    Correspondence: andrea.bassi@ke-srl.com

**Abstract:** The removal of fossil fuel subsidies and the introduction of carbon pricing have been discussed for more than a decade, but their potential contribution to emission reduction is still uncertain, especially in relation to the potential indirect impact of revenue recycling. We have created a simulation model, GSI-IF, designed to assess the emission reduction potential resulting from removing fossil fuel subsidies and recycling part of the avoided subsidy and additional revenue from carbon pricing to renewable energy and energy efficiency. Our results show that emissions could decline by 7.1% in 2030 and up to 19.8% in 2050 compared to a baseline scenario. We find that subsidy removal is most effective in reducing emissions in countries with a high incidence of fossil fuel subsidies and it has stronger impact in the short term. The recycling of carbon pricing is most relevant for larger economies and its impact accumulates over time, generating growing GHG reductions year after year. In the current context (year 2022) with high energy prices, heavy stress on fiscal balances, and the renewed ambition of most governments to reduce emissions toward Net Zero in 2050, subsidy removal and carbon pricing hold promise in the toolbox of decarbonization options while improving fiscal sustainability.

**Keywords:** fossil fuel subsidy; revenue recycling; energy model; forecasting; air emissions; system dynamics

## 1. Introduction

Over the last three decades, fossil fuel subsidy reform has been recognised as an integral tool to reduce GHG emissions. On the other hand, we find that the vast majority of studies that assessed the impact of subsidy removal on emissions did not consider the possibility to reallocate part of the subsidy savings to investments in energy efficiency and renewable energy in order to further reduce emissions. Furthermore, it is critical to distinguish between the impact of energy efficiency (which reduces the energy cost for households by lowering consumption and emissions) versus the impact of renewable energy (which reduces future costs, not current expenditure, by lowering the impact of future carbon pricing). There are differentiated impacts and they can offer insights to policymakers who are planning intervention options for the achievement of NDC targets. This research builds, and extends, on the expansive number of studies on the topic, varying in geographical scope, definition and calculation of subsidies, assumptions, sectors and fuels under consideration, reform period, type of subsidy, and the model leveraged for analysis [1].

With the current COVID-19 crisis, the need for fossil fuel subsidy reform is further exacerbated, with the burden of economic stimulation and socio-economic challenges falling on a governments' fiscal budget. As oil production recedes in response to the lowering demand, the pandemic also presents oil producing countries the opportunity to address fossil fuel subsidies.

In many instances, the removal of fossil fuel subsidies is projected to lead to a contraction of energy demand and a decline in emissions compared to a scenario with fossil fuel subsidies [2–5].

A variety of methodologies and models have been applied to analyzing questions surrounding fossil fuel subsidies, fossil fuel taxation, and their impact on energy demand and emissions, such as the World Energy Model [6], the EIRIN model [7], and the OECD-ENV Linkages model [8] to just name a few. The Global Subsidy Initiative (GSI) of the International Institute for Sustainable Development has developed the GSI Integrated Fiscal (GSI-IF) model to analyze the impacts of reforming fossil fuel subsidies, introducing carbon pricing, and repurposing subsidy savings and tax revenues on energy consumption and GHG emissions [9].

The GSI Integrated Fiscal (GSI-IF) Model was created to (1) support the estimation of the impact of fossil fuel pricing policy (subsidy removal and fuel taxation) on GHG emissions and (2) improve policy formulation and evaluation analysis for the elaboration of coherent and comprehensive climate mitigation strategies. The GSI-IF model forecasts energy demand at the national level by sector and source. This model, which uses social and economic drivers to determine future energy consumption and related GHG emissions, practically integrates sectoral data and knowledge from a variety of sources in one single model framework for a coherent analysis [9]. The second objective is met by allowing users to test several scenarios of fossil fuel subsidy removal across sectors and over time. Sensitivity analysis is also carried out to assess the degree of uncertainty surrounding model results when baseline assumptions are modified.

In this paper we present the latest findings that we produced using an updated version of GSI-IF in order to assess the extent to which removing fossil fuel subsidies, introducing carbon taxation, and reallocating part of the resulting revenues to energy efficiency and renewable energy can contribute to reducing emissions.

## 2. Literature Review

Several methods and models have been used to quantify the impact of fossil fuel subsidy removal and the introduction of fuel taxes on GHG emissions. These include sectoral energy models that focus on energy demand using a bottom-up (technology-rich) approach, macroeconomic models that estimate energy consumption using a top down (optimization of econometric) approach, or hybrid models that consider both technologies and macroeconomic dynamics. The following literature review focuses, first, on methods and models and, second, on the range of results generated by the studies published so far on this topic.

### 2.1. Methods and Models for Fossil Fuel Subsidy Removal

Four main methodologies applied to the modeling of fossil fuel subsidy reform emerge from our literature review: (i) partial equilibrium models (e.g., sectoral energy models), (ii) macro-econometric models, (iii) computable general equilibrium (CGE) models, and (iv) System Dynamics models. The following models are most commonly found in the literature surrounding questions of fossil fuel subsidy reform and taxation: (i) MARKAL-TIMES and the IEA's World Energy Model (WEM), which simulate detailed energy trends and their impact on socio-economic elements, including emissions at global, regional, and national levels using four scenarios; (ii) Cambridge Econometrics' global, which is a macro-econometric E3ME model that captures two-way linkages and feedbacks amongst economies, energy systems, emissions, and material demands; (iii) the OECD's ENV-Linkages, which is a global recursive-dynamic neo-classical CGE model calibrated to the ENV-Growth model that helps governments identify least-cost environmental policy options; and (iv) the GSI-IF model developed by the Global Subsidy Initiative of the International Institute for Sustainable Development, which is a system dynamics model calibrated to forecast national energy consumption and related emissions by fuel and sector for 26+ countries [3].

The IEA's WEM is a comprehensive energy demand model that covers energy developments in 25 regions up to the year 2040 [6]. The MARKet ALlocation (MARKAL) model is a PE model that optimizes energy supply to minimize production costs, or, more specifically, a "partial equilibrium bottom-up energy system technology optimization model employing perfect foresight and solved using linear programming" [10]. These models represent the entire energy chain from primary energy resources to energy service demands and operate under perfect foresight assumptions in order to optimize energy flows. Fossil fuel subsidies in these models are most often calculated using the price-gap approach. In other words, final end-user prices are compared to reference price and, if final end-user prices are lower than the reference price, the difference is assumed to be a subsidy. Subsidies are captured for both fossil fuels consumed by end-users and fossil fuels used as inputs for power generation [6]. Additional key inputs to the model are regional $CO_2$ prices that affect relative fuel costs and hence alter the composition of energy demand. The price of $CO_2$ is set to increase at varying rates between 2025 and 2040.

The E3ME model uses a macroeconometric modeling approach [11]. At their core, econometric models are top-down models that forecast the future based on past observations. Historical data is collected, and various statistical techniques are deployed, in order to estimate how a change in one variable is correlated to the change in another variable. The data on past correlations is then used to forecast the future development of a system. The model contains dynamic relationships, captures feedback effects, and is developed to forecast behavior based on empirical evidence (as opposed to energy-optimization models) [11]. The use of historical data to forecast the future behavior of systems lends itself well to short- or medium-term assessments related to issues for which sufficient historical data is available. Often however, we observe that long term trends exhibited by the structure of such models are unrealistic and that key constraints related to carrying capacity go unaccounted for. Additionally, given that future behavior is calibrated based on past data, results related to the assessment of policy impacts can be challenged [12]. The E3ME model is a model of the economy, energy systems, and environmental emissions that consists of 22 estimated sets of equations that are disaggregated by sector and country [11]. The structure disaggregated by sector allows for a relatively detailed estimation of fossil fuel subsidies by sector [13]. Furthermore, the E3ME model allows for the assessment of various policy interventions related to reforming fossil fuel subsidies, such as the analysis of economic and environmental impacts of environmental tax reform [14].

The OECD's ENV-Linkages model is a recursive dynamic neo-classical computable general equilibrium model (CGE) and successor to the OECD GREEN model [8]. CGE models are a standard tool for empirical analysis and they can be linked with energy models such as MARKAL or TIMES for analyzing energy demand and supply in more detail [10,15]. CGE models typically optimize for one specific actor in the system or one specific target indicator. The optimization feature is one of the main limitations of CGE models, as optimizing for one performance indicator neglects multiple trade-offs and may generate unrealistic development trajectories. This may lead to the generation of a scenario that optimizes parameters in order to achieve the most socially desirable outcome, but is inconsistent with actual past developments observed in reality [16]. The ENV-Linkages aims to provide information for policy makers concerning the phase out of fossil fuel subsidies and other green growth policies. Together with the PBL's IMAGE model, the ENV-Linkages model was used to generate the underlying database for the OECD Environmental Outlook to 2050, in which several policies, including fossil fuel subsidy reform, were analyzed in detail [17].

The GSI-IF model was developed back in the year 2014 to complement analyses conducted with the models above, and provide timely support, with a customized approach, to governments interested in exploring the inclusion of fossil fuel subsidy reform in their INDCs. The model supports the improved understanding of the impact of energy pricing, including fossil fuel subsidy removal, on energy conservation, fuel switching, and hence on emissions [18]. The GSI-IF model has been used for estimating the potential emission

reduction resulting from the recycling and repurposing of subsidy savings and fuel tax revenues into energy efficiency and renewable energy, as presented in the next sections.

*2.2. Review of Impacts Attributable to Fossil Fuel Subsidy Removal and Taxation*

A well-established body of literature is available on the impacts of reforming fossil fuel subsidies. Information is available both from scientific studies and from past experience with reforming fossil fuel subsidies. The literature indicates that fossil fuel subsidy reform is projected to reduce global emissions by 1 per cent to 10 per cent by 2030, depending on the base year, fuels covered, scope of subsidy, and methodology used. This further increases when fossil fuel subsidy reform is integrated with other fiscal tools and efficiency measures. For instance, an IEA study estimates 37 per cent emission reductions by 2040 globally compared to the IEA New Policy Scenario when a subsidy phaseout in oil exporting countries is combined with energy efficiency measures [19]. According to the latest report, emissions are projected to decrease by 700 Mt by 2030 [4]. Nationally, emissions are projected to reduce significantly in some regions by 2050 from subsidy removal solely, such as in Russia, India and China by about 25 per cent, 18 per cent and 8 per cent, respectively [20]. However, they are also projected to rise in others, such as Japan and the U.S., where emissions increase by about 9 per cent and 7 per cent, respectively (Ibid.).

One of the major risks of fossil fuel subsidy reform are unintended side-effects that counteract the envisaged emission reduction targets. For example, in countries in which GHG emissions are high, cheaper coal is readily available to replace lesser emission intensive oil and gas. The consequence of a global subsidy removal could hence increase emissions in the medium or long term as oil and gas become more expensive and are substituted for coal [21,22]. For instance, though emissions reduced due to a fossil fuel subsidy removal in China from 2003 to 2014, it also led to substituting oil and gas with high-carbon coal, which increased emissions in later years; this highlights the need for additional, long-run policies and measures that avoid replacement of fossil fuels with even higher carbon technologies, e.g., increasing the share of renewables in the energy mix and raising finance for such activities [23]. Furthermore, the mitigation effect from production subsidy removal is maximized only when subsidies are removed for all fossil fuels, which avoids production to move toward the most subsidized fuel, arising from partial substitution [24].

In addition to the reform of fossil fuel subsidies, emission reductions can also be attributed to fossil fuel taxes, whether or not combined with subsidy reform. For instance, a study assessing the impact of carbon-taxed fuels on emissions in British Columbia found that emissions reduced by 10 per cent between 2008 to 2011, overtaking the reduction in other regions in Canada by 9 per cent [25]. Similarly, another study found that Sweden's carbon tax reduced emissions by 6 per cent in an average year, while another 5 per cent reduction could be attributed to the value-added tax [26]. This study also highlights that the carbon-tax elasticity of demand for the transport fuel (gasoline) is thrice the price elasticity—using price elasticity to project emission reductions could thus be inaccurate. Combining a carbon tax with a value-added tax, goods and services tax, and producer tax could help correct the fossil fuel prices. In line with these findings, Sterner amplifies the example of European countries that introduced high transport fuel taxes and cut emissions by over 50 per cent from 1978 to 2003, leading to a long term impact of 10 per cent emission reductions from all sources globally [27].

Research has found that production taxes, particularly on thermal coal, have a significant impact on emission reductions; a global tax of USD 10/t $CO_2$ on thermal coal production can reduce global emissions in the electricity sector by more than 19 per cent (1.9 Gt $CO_2$) [28]. The same study also found that production taxes are not only a significant and reliable method for curbing coal use globally, but also raising tax revenues as compared to unilateral export taxes. Curbing coal use is integral to meeting the Paris Agreement long-term temperature goals; more than 80 per cent of current coal reserves must remain unused if we are to do so [29].

The literature provides a key solution to these issues: subsidy "swaps" that combine subsidy removal with the implementation of clean energy support measures. For instance, employing cost-effective industrial energy efficiency measures could lead to additional emission reductions of about 2 Gt $CO_2$ in 2030 [4]. This is supplemented by a study in China that found that emissions reduced by an additional 15.5–19.1 per cent when fossil fuel subsidies were swapped with clean energy support measures as compared to when fossil fuel subsidy reform alone took place [30]. Another example found that investing revenues from subsidy removal in the EU into renewable energy (solar photovoltaic) installations increased estimated emission reductions from 1.8 to 2.2. per cent by 2030 [31].

It is thus established that fossil fuel subsidy reform or removal alone will not be sufficient for a long-term, sustained, reduction in emissions. As such, countries will need to (i) phase out their fossil fuel subsidies, (ii) correct for undermined fossil fuel prices through appropriate taxation, which could reduce global emissions by an increased 28 per cent [32], and (iii) adopt clean energy support measures to shift to low carbon fuels [33]. By doing so, governments are able to achieve emission reductions while simultaneously both saving government resources and raising revenue—both results are integral to a national pandemic response and recovery measures. This is corroborated by Monasterolo and Raberto (2019) [7] who show that subsidy reform improves macroeconomic performance by enabling increased capital accumulation domestically, while also creating employment and investments [7]. In order to avoid carbon lock-ins, these considerations are amplified in the context of the COVID-19 pandemic. As such, in addition to these measures, countries must also (i) align their short- and medium-term clean energy support measures in response to the pandemic [34] and(ii) design green stimulus package evaluation before and after implementation [35]. As of November 2021, governments in the G20 countries have committed at least USD 318.83 billion in support of fossil fuel energy and USD 279.26 billion in support of clean energy, again highlighting the predominance of fossil fuels in developed economies and their recovery from the pandemic [36].

## 3. Materials and Methods

### 3.1. Introduction to the Global Subsidy Initiative—Integrated Fiscal (GSI-IF) Model

The GSI-IF model is a System Dynamics-based simulation model developed to forecast energy demand and related emissions for a number of countries. The model projects national energy consumption by sector and source from 1990 to 2040. GHG emissions are estimated based on the Energy demand projections generated by the model and IPCC GHG multipliers by fuel source [37]. The model uses continuous-time simulation to generate 'what if' scenarios. The model was designed to be customized to a new country in one to two days, and for simulations to run in seconds, so that these can be used to inform discussions around the impacts of fossil fuel subsidy phase-out and the introduction of carbon taxes on total GHG emissions, directly in in-person and remote meetings [3,18,38]. The setup of the model allows the simultaneous implementation of measures as well as for phasing of measures, such as, for example, the removal of subsidies between 2020 and 2025, followed by the introduction of a carbon tax between 2025 and 2030.

The GSI-IF Model is characterized by the following features:

- Boundaries: the model focuses on energy consumption, and it does not include emissions from other sectors (e.g., land cover). Furthermore, the model does not include an endogenous estimation of energy supply. On the other hand, users can modify the energy mix for electricity generation, which is also based on existing national plans and on possible reuse of policy-induced subsidy savings.
- Granularity: The model is customized to represent national energy consumption, and it is not disaggregated spatially at the sub-national level. On the other hand, the model includes energy consumption from the residential, commercial, industrial, and transport sectors, which is disaggregated into coal, petroleum products, natural gas, biofuels and waste, and electricity.

- Time horizon: the GSI-IF Model is built to analyze medium to long-term trends. Simulations start in 1990 and extend up to 2040. Starting in 1990 supports both model validation and the correct assessment of long-term trends, including the identification of possible underlying socio-economic structural changes.
- Structure: the GSI-IF Model is relatively small, and it uses the following key exogenous drivers: GDP, population, energy efficiency (as an annual percent increase), and energy prices. Users can also modify the energy mix for electricity generation, change default values for price and income elasticity, as well as emission factors.
- Dataset: several data series and data inputs are required to customize and simulate the GSI-IF Model. The current version of the model uses energy consumption and electricity supply data from IEA's World Energy Balances; GDP and population are either available from the IEA or from World Bank's World Development Indicators (WDI) and the UN World Population Prospects database; energy prices were primarily collected from national government sources; fossil fuel subsidies figures were extracted from GSI's database, which includes estimates from the IEA, OECD and IMF; the cost of renewable energy (RE) electricity generation capacity and energy efficiency (EE) interventions are obtained from the IEA; price and income elasticities were determined based on literature review and model calibration to historical trends.

One of the main drivers of the model is energy price, which can be modified in two ways: (1) by setting baseline medium to longer-term trends and (2) by removing fossil fuel subsidies and introducing fuel and/or carbon taxes. By setting a range of scenarios with varying medium- to longer-term trends for energy prices, the model enables the analysis of energy consumption projections considering different underlying global energy price scenarios. Changes in energy price will affect fuel substitution and hence alter sectoral energy demand and energy mix. The removal of subsidies increases energy prices, which lowers energy demand in two possible ways: energy is becoming more expensive and consumption is reduced to offset the growth in expenditure, and, if energy services are required, the use of (previously) subsidized fossil fuels declines and consumption of now comparatively cheaper fuels increases. GHG emissions are affected by both the reduction in energy consumption and the change in fuel mix, and the GSI-IF Model analyses these effects separately. Furthermore, in addition to the removal of subsidies, the GSI-IF Model estimates the annual subsidy savings and provides the option to reallocate these savings to additional energy efficiency or to increase the share of renewable energy in the electricity mix, or both. As a result, the GSI-IF Model allows us to estimate the impact of fossil fuel subsidy removal on GHG emissions, and compare such reduction to other possible intervention options (e.g., investments and/or mandates on energy efficiency and renewable energy). Emission reductions can also be estimated as a result of the reallocation of subsidy savings, such as through investments in additional renewable energy generation capacity and energy efficiency improvements beyond the baseline.

Finally, the GSI-IF Model complements other existing models and methodologies for the estimation of emission reductions from the implementation of various policy interventions across sectors. To provide two examples, the GSI-IF Model can be used as an input to other energy models, such as MARKAL, TIMES, and LEAP, as it estimates demand for energy services using GDP, population, and energy prices as inputs; also, the GSI-IF Model complements the IPCC Inventory Software, which helps users to compile their national greenhouse gas inventory, with the generation of projections for energy demand and with the assessment of the outcomes of (selected) policy implementation.

### 3.2. Energy Demand Modeling in GSI-IF

The national energy consumption forecasts generated by the GSI-IF Model are disaggregated by fuel source and sector. Specifically, the model includes energy consumption from coal, petroleum products, natural gas, biofuels and waste, and electricity. The consumption of each fuel source is further disaggregated into sectoral energy consumption, specifically residential, commercial, industrial, and transport sectors. The number of coun-

tries in the GSI-IF Model has been updated compared to earlier iterations and is now available for 32 countries.

The main structural assumptions of the model are (see Figure 1):

- Final energy consumption is estimated considering (1) indicated demand (including the effect of GDP, population, and energy efficiency); (2) the price effect; and (3) the substitution effect. Items (1) and (2) are used to estimate demand for energy services.
- The potential for fuel substitution is represented by the ratio of an energy price over the national weighted average energy price. This implies that an energy source will become more attractive if its price increases less than others when subsidies are removed.
- It is assumed that price effects require a 1-year delay to influence energy consumption.

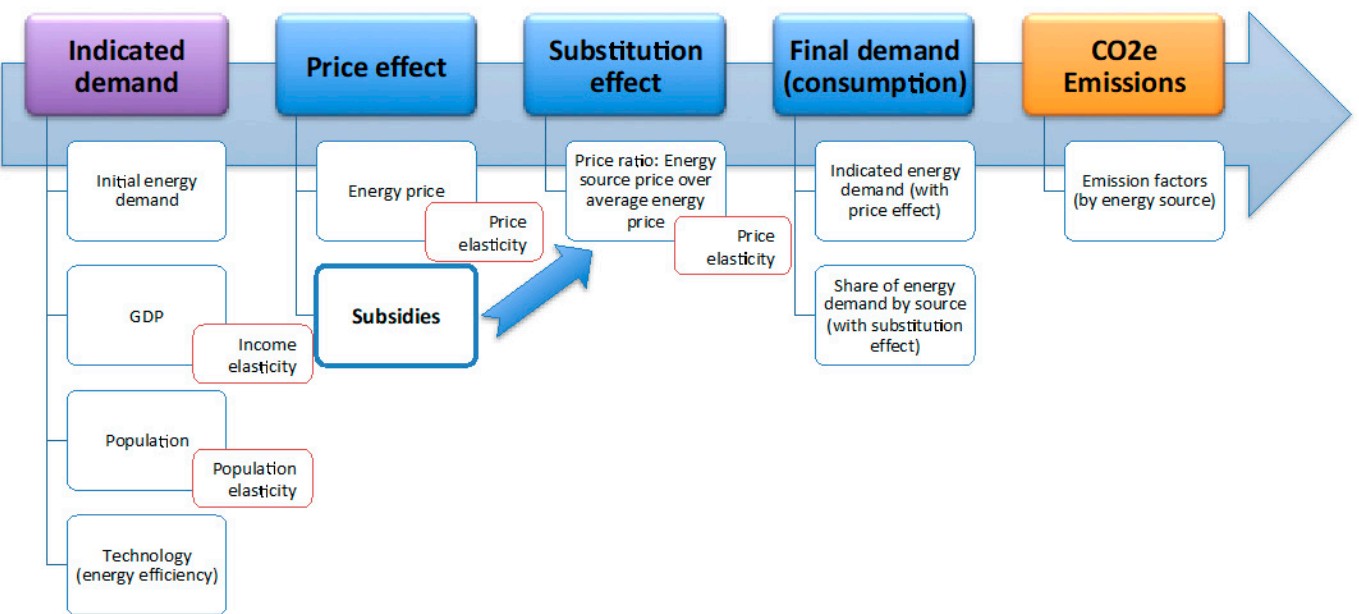

**Figure 1.** GSI-IF model sketch, highlighting the main steps considered for estimating $CO_2$e emissions reductions resulting from FFSR.

The model considers several "energy demand" variables considering the effects of price and substitution as indicated in Figure 1. This section provides a description of how indicated sectoral energy demand is calculated (coal is presented as an example) using the initial value for 1990, multiplying it an index for GDP and population (both indexed to 1990) and dividing it by relative energy efficiency (also indexed to 1990). The use of subscripts, 'sector' in the example provided below, allows for calculating energy demand for the residential, commercial, industrial, and transport sectors within the same variable.

$$\text{Indicated Coal Demand[sector]} =$$
$$\frac{(\text{initial coal demand table[sector]} \times \text{relative GDP}^{\wedge}\text{elasticity of coal demand to GDP[sector]} \times \text{relative population}^{\wedge}\text{elasticity of coal demand to population)}}{\text{relative energy efficiency}} \qquad (1)$$

The price effect is then added, taking indicated demand (presented above) and multiplying it by relative energy price (indexed to 1990) and raising it to the power of a price elasticity.

The removal of fossil fuel subsidies is reflected in energy price changes. When subsidies are removed, it is assumed that energy prices increase for all sectors (unless it is known that subsidies are allocated to specific users). Scenarios are customized by defining

the extent to which subsidies are removed and the timeline (e.g., full removal, linearly, by 2025).

$$\text{"Indicated Coal Demand (With Price Effect)"[country,sector]} =$$
$$\text{indicated coal demand[country,sector]} \times \text{relative coal price[country]} \hat{} \text{ elasticity} \quad (2)$$
$$\text{of coal demand to coal price[country,sector]}$$

Subsequently, the potential for energy substitution is considered. The formulation is the same as the one used for incorporating the price effect, but a delay of 1 year is used to represent the lag existing between price changes and demand (or consumption) changes.

$$\text{"Coal Demand (With Substitution Effect)"[country,sector]} =$$
$$\text{delay n(("indicated coal demand (with price effect)"[country,sector]} \times \text{"coal}$$
$$\text{price-substitution"[country]} \hat{} \text{ elasticity of coal demand to coal price[country,sector]),}$$
$$\text{time to adapt demand to price changes, ("indicated coal} \quad (3)$$
$$\text{demand (with price effect)"[country,sector]} \times \text{"coal price-substitution"}$$
$$\text{[country]} \hat{} \text{ elasticity of coal demand to coal price[country,sector]), 3)}$$

The potential for substitution from one energy source to the other, due to price changes (e.g., as a result to fossil fuel subsidy removal), is incorporated here by using the ratio of energy source price (e.g., coal) over the average energy price of the country (estimated as a weighted average of all energy prices). This ratio represents the relative affordability of one energy source over the others, and it is also indexed to ensure consistency with the use of elasticities.

$$\text{"Coal Price-Substitution"[country]} =$$
$$\text{delay n((relative coal price[country]/relative weighted average energy} \quad (4)$$
$$\text{price[country]),1,1,1)}$$

Indicated energy demand (including the price effect) is used to estimate the total indicated energy demand (which is also a demand for energy services). The potential for substitution is instead used to estimate the actual share of energy consumption by energy source. As a result, a normalization is performed to consider indicated demand as well as the energy mix resulting from the full price effects (including substitution).

$$\text{Normalized Coal Demand[country,sector]} =$$
$$\text{total indicated country energy demand[country]} \times \text{normalized coal share of} \quad (5)$$
$$\text{energy demand[country,sector]}$$

GHG emissions from energy consumption are calculated by sector and energy source using IPCC emission factors [37]. The normalized sectoral energy demand, by fuel, is multiplied by the respective emission factor.

$$\text{Coal emissions[country,sector]} =$$
$$\text{normalized coal demand[country,sector]} \times CO_2\text{e emissions per TJ of coal} \quad (6)$$

Emissions per TJ of electricity are calculated based on the share of oil, the share of coal, and the share of gas in the total electricity generation mix, all obtained from past data. Nuclear and renewable energy are assumed to have zero carbon intensity per TJ generated. A weighted average emissions per TJ of electricity multiplier is calculated based on the shares of oil, coal, and gas in total power generation and a respective emission factor.

$$\text{average emissions per tj of electricity generation} =$$
$$\text{((emissions per GWh of gas power generation} \times \text{share of gas[country](time)}$$
$$\text{+ emissions per GWh of coal power generation} \times \text{max(0, (share of oil and}$$
$$\text{coal[country]} - \text{share of oil[country](time)))} + \text{emissions per GWh of liquid} \quad (7)$$
$$\text{fuel power generation} \times \text{share of oil[country](time))/conversion from GWh}$$
$$\text{to TJ)}$$

The recycling and repurposing of subsidy savings and tax revenues is performed endogenously by estimating the additional energy efficiency potential and additional renewable energy capacity based on annual fossil fuel subsidy savings and additional tax revenues. The GSI-IF model allows for reallocating a specific percentage of subsidy savings to energy efficiency and/or renewable energy, as well as to household compensation, and impacts are reflected in altered behavior for energy efficiency and the GHG intensity per TJ of electricity. Data required for estimating the impact of revenue recycling are the cost of energy efficiency (per TJ of energy saved) and the cost of renewable generation capacity (per MW of capacity).

Both increased energy efficiency and additional electricity generation from renewable energy feed back to the estimation of (i) indicated demand (in the case of energy efficiency) and (ii) emissions from power generation (in the case of renewable energy). In other words, the improvement of energy efficiency partly or fully offsets the cost increase resulting from subsidy removal and fuel taxation, while reducing emissions; in contrast, renewable energy reduces the carbon intensity of power generation, but also reduces emissions.

## 4. Results

Four scenarios were simulated with GSI-IF for generating the results: business as usual (BAU), Low ambition, Medium ambition, and High ambition. Fossil fuel subsidy removal (FFSR) and fuel tax (FFSR + Tax) are included in all low carbon scenarios, with a varying level of ambition. The BAU scenario represents the baseline for assessing the impacts of fossil fuel subsidy removal and fossil fuel taxation on the forecasted GHG emissions of the 32 countries considered. Key assumptions for the three scenarios are summarized in Table 1.

**Table 1.** Documentation of scenario assumptions.

| Scenario | FFSR | | Energy Tax | |
|---|---|---|---|---|
| | **2025** | **2030** | **2025** | **2030** |
| BAU | N/A | N/A | N/A | N/A |
| Low ambition | Full removal by 2030 (linear) | | N/A | 10% |
| Medium ambition | Full removal (linear) | Fully removed | N/A | 20% |
| High ambition | Full removal (linear) | Fully removed | N/A | 30% |
| **Reallocation** | | | | |
| Reallocation to EE | 20% | | | |
| Reallocation to RE | 10% | | | |

In the BAU scenario, total energy demand for all countries continues to grow unaffected by policy decisions. The phase-out of fossil fuel subsidies, the introduction of a carbon tax, and resulting reallocation of savings to energy efficiency in the three policy scenarios cause energy demand to contract in the short- and medium-term, with a stronger impact emerging over time. Between 2020 and 2030, total energy demand is on average 4.7% lower in the Low ambition scenario, 6.6% lower in the Medium ambition scenario, and 7.1% in the High ambition scenario, respectively. Between 2020 and 2050, the average reductions in total energy demand compared to the BAU scenario are 12.7%, 16.7% and 19.8% for the Low, Medium and High ambition scenarios, respectively.

The phase-out of subsidies and the implementation of an additional carbon tax contributes to emission reductions compared to the baseline scenario in two ways: (i) by affecting energy prices and hence the energy mix for the countries considered, and (ii) through the reallocation mechanism, which reduces emissions through increasing energy efficiency and the share of renewables in total power generation. Figure 2 presents the forecasted reduction in GHG emissions in the High ambition scenario compared to the baseline scenario and illustrates the shares contributed by (i) and (ii), respectively, for the year 2030. The

results indicated that, with few exceptions, renewable energy and energy efficiency are the main drivers for emission reduction in most countries. Countries in which the phase-out of fossil fuel subsidies is the dominant force for emission reductions are Venezuela, Iraq, Egypt and Zambia, or the countries with largest proportional share of subsidies in GDP.

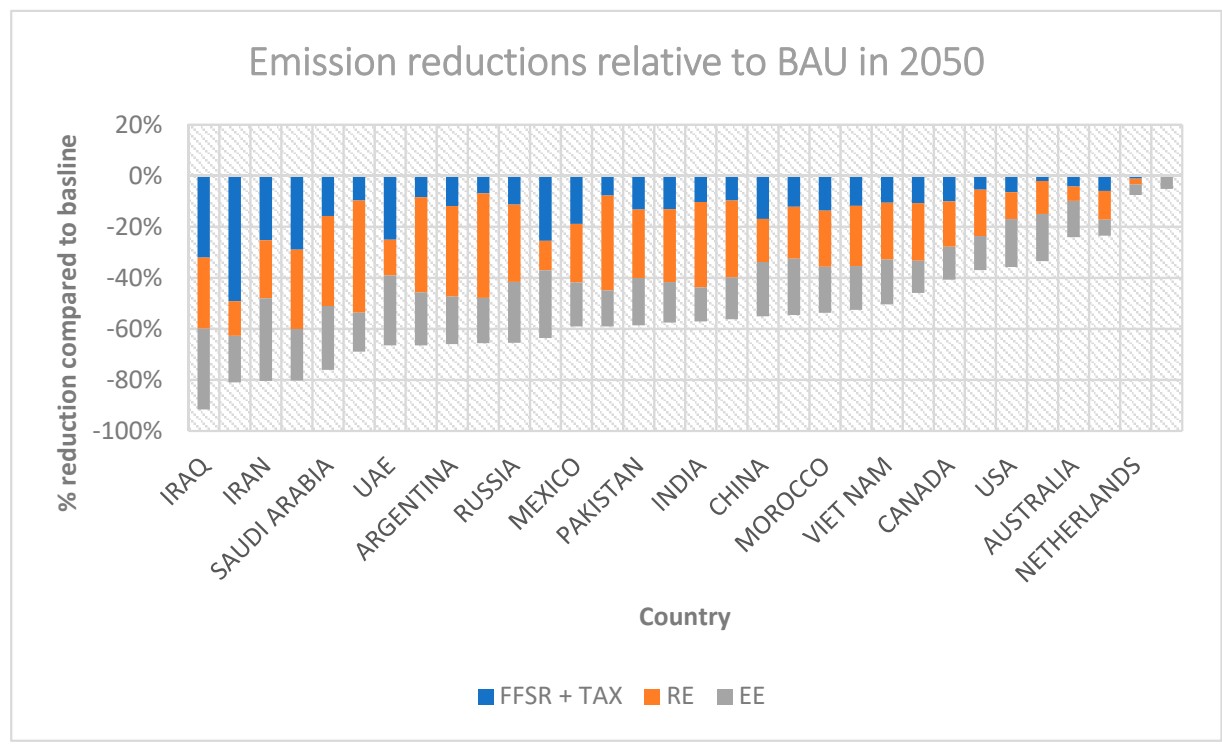

**Figure 2.** Projected percent reduction in GHG emissions relative to BAU emissions in 2050 (High ambition scenario).

In the Low ambition scenario, the linear phase-out of subsidies between 2020 and 2030 combined with a 10% tax is projected to reduce GHG emissions between 2020 and 2030 on average by 7% across all countries. In the longer term, energy-related emissions are projected to decline on average by 14.8% (2020–2040) and 25.3% (2020–2050), respectively. The total indicated reductions in GHG emissions for the Low ambition scenario are equivalent to annual reductions of 1.54 Gt (2020–2030), 5.12 Gt (2020–2040), and 13.2 Gt (2020–2050) per year compared to the BAU scenario.

The more ambitious trajectory for the phase out of subsidies and carbon taxes reduces emissions between 2020 and 2030 on average by 10.1% (Medium ambition) and 10.8% (High ambition) compared to the BAU, respectively. In fact, the High ambition scenario exhibits almost double the reductions observed for the Low ambition scenario over the first decade. The indicated reductions correspond to avoided emissions of 2.46 Gt per year (Medium ambition) and 2.88 Gt (High ambition), respectively. Total reductions in annual GHG emissions for the period 2020 to 2040 are projected between 7.99 Gt (Medium ambition, −19.9% vs. BAU) and 10.33 Gt per year (High ambition, −22.9% vs. BAU), and the long-term average reduction ranges from 32.9% in the Medium ambition scenario (−20.25 Gt per year vs. BAU) to 38% in the High ambition scenario (−26.13 Gt per year vs. BAU).

The impact of reforming fossil fuel subsidies varies by country, depending on the current fuel mix, the type, and the quantity of subsidies in place. Countries with a high subsidy to GDP ratio, such as Venezuela, Iran, Egypt, Zambia and Algeria, see the strongest relative contractions in energy demand and are among the countries where the highest relative reductions in emissions are observed (see Figure 2). In the High ambition scenario, the reduction in emissions averages 26.1% (2020–2030) and 52.1% (2020–2050), respectively,

which is around 15.3% and 14.1% higher, respectively, compared to the average across all countries. The phase out of fossil fuel subsidies leads to large price changes in those countries, which causes energy conservation and fuel switching.

The size of the economy and energy demand is what matters the most when it comes to the potential emission reduction from energy efficiency and renewable energy. For instance, the average share of subsidy savings over GDP for China, USA, India, and Indonesia averages around 1.3% between 2020 and 2050, while emission reductions from those four economies alone constitute between 61.3% (2020–2030) and 78.5% (2020–2050) of total emission reductions forecasted for all 32 countries. Additional investments in energy efficiency and renewable energy have a much stronger impact on emission reductions in these economies, as, despite the small average share of subsidy savings over GDP, the amount of resources reallocated in form of subsidy savings and fossil fuel taxes is much higher compared to smaller economies. Absolute emission reductions in the High ambition scenario are presented in Figure 3 for all countries. Figure 3 ranks countries according to absolute emission reductions compared to the baseline in the year 2050. The results show that the highest reduction in emissions from the phase-out of fossil fuel subsidies is forecasted for China, India, USA, and Indonesia.

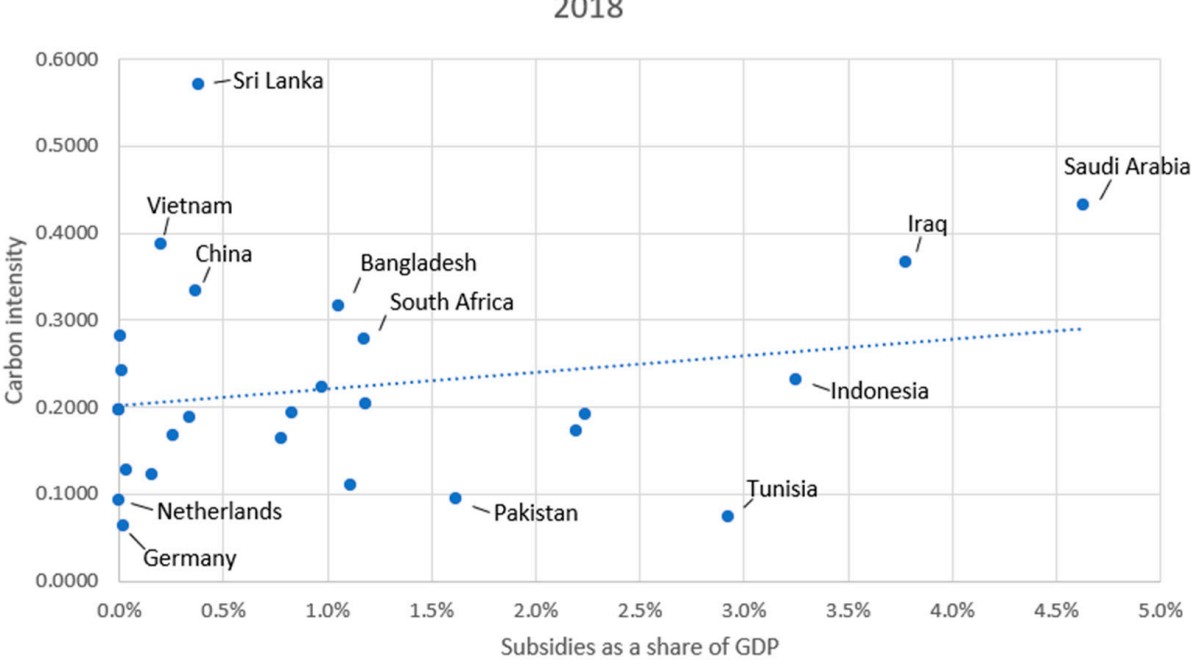

**Figure 3.** Carbon intensity vs. subsidies as share of GDP, year 2018 (Source IMF and IEA).

Table 2 presents GSI-IF results for annually avoided emissions in the year 2050 in Mt, total, and by intervention. The cumulative percent reduction potential highlights that, based on our projections, 50% of total reductions will be achieved in China, while another 30% of reductions will be realized in India, USA, and Indonesia. The countries in Table 2 are ranked according to potential emission reductions.

In summary, the scenarios simulated indicate that the phase-out of fossil fuel subsidies in combination with fuel taxes affects country-specific energy demand dynamics and contributes to reducing GHG emissions. For the period from 2020 to 2030, cumulative reductions range from 16.89 Gt (Low ambition scenario) to 31.71 Gt (High ambition scenario). Over the lifetime of investments, considering the period from 2020 to 2050, the projections indicate cumulative avoided emissions of 281.58 Gt in the Low ambition scenario, 436.67 Gt in the Medium ambition scenario, and 569.26 Gt in the High ambition scenario. On average, the reduction forecasted is equivalent to 9.39 Gt, 14.56 Gt, and 18.98 Gt of avoided emissions per year in the Low, Medium, and High ambition scenario over the next 30 years.

**Table 2.** Forecasted emission reductions in Mt per year High ambition scenario relative to BAU—by country and intervention—year 2050.

| Country | FFSR + TAX | RE | EE | Total | % of Total Reduction Potential | Cumulative Total % Reduction Potential Across Countries |
|---|---|---|---|---|---|---|
| China | −4417 | −9323 | −7349 | −21,089 | 50.06% | 50.06% |
| India | −1425 | −2851 | −2070 | −6346 | 15.06% | 65.13% |
| Usa | −542 | −2573 | −981 | −4095 | 9.72% | 74.85% |
| Indonesia | −431 | −966 | −683 | −2080 | 4.94% | 79.79% |
| Saudi Arabia | −325 | −348 | −227 | −900 | 2.14% | 81.92% |
| Japan | −122 | −549 | −191 | −862 | 2.05% | 83.97% |
| Viet Nam | −212 | −374 | −273 | −859 | 2.04% | 86.01% |
| Russia | −156 | −341 | −189 | −687 | 1.63% | 87.64% |
| Egypt | −213 | −191 | −272 | −676 | 1.61% | 89.24% |
| Germany | −66 | −389 | −169 | −624 | 1.48% | 90.72% |
| South Africa | −107 | −217 | −150 | −474 | 1.12% | 91.85% |
| Pakistan | −109 | −228 | −128 | −465 | 1.10% | 92.95% |
| Australia | −80 | −249 | −135 | −464 | 1.10% | 94.05% |
| Bangladesh | −136 | −165 | −124 | −425 | 1.01% | 95.06% |
| Mexico | −66 | −177 | −139 | −382 | 0.91% | 95.97% |
| Iran | −115 | −64 | −126 | −305 | 0.72% | 96.69% |
| Canada | −38 | −127 | −93 | −257 | 0.61% | 97.30% |
| Brazil | −15 | −95 | −136 | −246 | 0.59% | 97.89% |
| Morocco | −27 | −81 | −42 | −150 | 0.36% | 98.25% |
| Uae | −37 | −36 | −46 | −119 | 0.28% | 98.53% |
| Netherlands | −14 | −62 | −35 | −111 | 0.26% | 98.79% |
| Argentina | −24 | −39 | −32 | −96 | 0.23% | 99.02% |
| Iraq | −56 | −16 | −21 | −93 | 0.22% | 99.24% |
| Algeria | −30 | −14 | −31 | −75 | 0.18% | 99.42% |
| Sri Lanka | −12 | −38 | −15 | −64 | 0.15% | 99.57% |
| Nigeria | −10 | −18 | −35 | −63 | 0.15% | 99.72% |
| Myanmar | −7 | −14 | −8 | −28 | 0.07% | 99.79% |
| Ghana | −4 | −7 | −12 | −24 | 0.06% | 99.84% |
| Venezuela | −7 | −6 | −7 | −21 | 0.05% | 99.89% |
| Tunisia | −4 | −7 | −8 | −19 | 0.05% | 99.94% |
| Ethiopia | 2 | −1 | −15 | −14 | 0.03% | 99.97% |
| Zambia | −2 | −3 | −7 | −12 | 0.03% | 100.00% |
| Total | −8808 | −19,570 | −13,748 | −42,125 | 100.00% | |

## 5. Discussion

The data collected to parametrize and customize GSI-IF indicate that the lower the carbon intensity of an economy, the lower the share of subsidies over GDP (see Figure 3). Practically, fossil fuel subsidies create a market distortion that prevents countries from moving toward decarbonization.

Our results indicate that the removal of fossil fuel subsidies and the introduction of carbon taxation can trigger a virtuous mechanism when revenue recycling is introduced

in favor of investments in energy efficiency and renewable energy. Countries with high subsidies relative to GDP and high carbon intensity, such as Saudi Arabia and Iraq, hold significant potential for fossil fuel subsidy reform and emission reduction. We assume that only 30% of the revenue generated can be recycled to stimulate further investment, leaving ample resources for the mitigation of short term undesirable economic consequences of carbon pricing measures. In countries with large economies, the potential lies in triggering investments, rather than on removing subsidies also. In China, USA, and India the size of the economy allows governments to better leverage the lasting impact of carbon pricing in order to obtain long term sizable emission reduction.

Our results may seem striking in terms of potential emission reduction. We have highlighted the short-term role of subsidy removal, coupled with the compounding effect of investments in energy efficiency and renewable energy that happen every year, until 2050. The literature shows that our results are aligned with other studies. For instance, the IEA has modeled the phase-out of subsidies by 2030 in the context of sustainable recovery scenarios, and reports that global $CO_2$ emissions would be around 700 Mt lower relative to a scenario where subsidies would remain in place [4]. Furthermore, the more recent observations reported for the introduction of a fossil fuel tax suggest that a 10% increase in price leads to reductions in energy use and related emission in the range of 5.2% and 9% [5], which is consistent with the findings reported for the low ambition scenario in this study. In general, the 2030 reductions generated by the GSI-IF model are consistent with the wider literature on fossil fuel subsidy reform [2]. Furthermore, the IMF estimates that removing subsidies has the potential to reduce emissions of up to 28 per cent in 2015 [32,39–41]. In contrast, a study using the CGE model that assesses the removal of both coal producer and consumer subsidies in OECD and non-OECD countries finds that this leads to 8 per cent fewer emissions than what would have arisen early in the 21st century [42]. Another study shows that attaching a price of EUR 30 and EUR 60 per ton of $CO_2$ has the potential to reduce emissions by up to 30 per cent and 40 per cent, respectively, by 2030 [43].

When interpreting these results, it should also be considered that the price of low carbon technologies is rapidly declining and that their adoption is likely to accelerate in the coming years due international commitments to emission reduction [44,45]. The adoption of such technologies may affect the business-as-usual scenario by reducing emissions, and hence potentially reduce the net decline in emissions that is achieved with subsidy removal and carbon pricing. Practically, if technologies are already becoming more economically viable, the price signal sent by subsidy removal and carbon pricing may become less effective in determining the uptake of new technologies and practices, but it would nevertheless support this process.

A consideration has to be made in relation to energy affordability when removing fossil fuel subsidies and introducing carbon pricing. First, the scenarios of reallocation of subsidy savings only consider that 30% of this amount will be invested in energy efficiency and renewable energy. The remaining 70% could be used to support low-income families, or the reduction in public deficit. Second, while the GSI-IF model does not estimate the impact of energy prices on energy spending, and hence on affordability, it estimates changes in energy consumption as a result of price changes. The change in energy consumption could be considered a proxy for energy affordability. Overall, we find that the introduction of subsidy removal and carbon taxes increase energy prices, but energy efficiency reduces consumption by an amount larger than the price increase. Over time, the affordability of energy would increase as a result (i.e., if energy consumption declines more than the increase in price, affordability will increase), but it is important to plan a phased removal of subsidies and a simultaneous improvement in energy efficiency.

While holding considerable potential, fossil fuel removal alone will not be sufficient to achieve the long-term Paris Agreement climate goals. We must understand that (i) fossil fuel subsidies are not the only source of emissions—there also exist other significant drivers such as population, economic growth, and resource depletion [21], and (ii) fuel substitution in favour of more emission-intensive fuels could also occur, such as in countries

where the availability of lower-emission fuels are low. Lastly, there is a risk of carbon leakage across borders when one country's or region's climate policies are stronger than another's—highlighting the need for coordinated measures. A study found that a global fossil fuel subsidy removal could lead to emission increases instigated by higher economic growth and international trade in countries that did not reform their subsidy policies [46]. Another example of cross-border emission leakage can be seen in a study by Buriaux, Chateau, and Savage, where consumption subsidy removal led to trade inflows in OECD countries as a result of a reduction in trade volumes in oil exporting countries [47].

## 6. Conclusions and Policy Implications

The removal of fossil fuel subsidies holds considerable potential for reducing emissions according to most of the studies conducted to date. The results of this study go one step further, nuancing country-specific differences when it comes to the reinvestment of subsidies in energy efficiency and renewable energy, with an eye toward deeper decarbonization and net zero targets by 2050. We find that the potential reduction in emissions emerging from the reallocation of 30% of subsidy savings to energy efficiency and renewable energy is potentially higher than the reduction caused by subsidy removal and carbon pricing, especially in countries characterized by high energy demand. This indicates a strong synergy with other climate mitigation options and an important cost-reduction potential in the face of possibly higher energy prices.

GSI-IF projections over the lifetime of renewable energy and energy efficiency investments, made possible by revenue recycling from fossil fuel subsidy removal and carbon taxation, considering the period from 2020 to 2050, indicate cumulative avoided emissions between 281.58 Gt in the Low ambition scenario and 569.26 Gt in the High ambition scenario, equivalent to almost 10 to 20 Gt of emissions per year on average.

This study highlights that the phasing out of fossil fuel subsidies does not only hold considerable savings potential for governments but also opens the doors for their redistribution toward sustainable development. The reinvestment of subsidy savings in renewable energy and energy efficiency (30% of the total amount of avoided costs and revenues generated) serves as an enabler for the emergence of more significant emission reductions via energy efficiency and renewable energy. Given the inefficient nature of most fossil fuel subsidy schemes, governments should seize the opportunity of reforming support to the energy sector, especially in light of the health cost caused by the use of fossil fuels.

While the policy-induced increase in energy prices may lead to a temporary contraction in economic activity and energy demand, lower energy spending, health costs and increased job creation are expected to more than offset potential short-term undesirable outcomes.

**Author Contributions:** Conceptualization, A.M.B. and R.B.; methodology, A.M.B. and G.P.; software, A.M.B. and G.P.; literature review, K.B. and G.P.; writing—original draft preparation, A.M.B., K.B., G.P. and R.B.; writing—review and editing, A.M.B.; project administration, R.B. All authors have read and agreed to the published version of the manuscript.

**Funding:** This research was funded by the Nordic Council of Ministers, grant number 62001ERTQ5.

**Institutional Review Board Statement:** Not applicable.

**Informed Consent Statement:** Not applicable.

**Data Availability Statement:** Data on subsidies were collected from available databased of the IEA, IMF and OECD. Energy demand data were collected from IEA national energy balances.

**Conflicts of Interest:** The authors declare no conflict of interest.

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
