# Peer review of "Emission Reduction via Fossil Fuel Subsidy Removal and Carbon Pricing, Creating Synergies with Revenue Recycling"

_world, doi:10.3390/world4020016_

Round 1

Reviewer 1 Report

Removing the fossil fuel subsidies and imposing the carbon pricing have been discussed for more than a decade, but their potential on the reduction of carbon emission is still unclear. In this work, the simulation model, namely GSI-IF, was designed to assess the emission reduction potential resulting from removing fossil fuel subsidies and recycling part of the avoided subsidy and additional revenue from carbon pricing to renewable energy and energy efficiency. In my opinion, the work is suitable for publication in World. However, the following remarks are carefully addressed.

(1)   What is the difference between this work and the previous study? And what is the practical significance of this work?

(2)   The modeling process and the structure of GSI-IF model should be specifically described.

(3)   What is the relationship between the removal of the fossil fuel subsidies as well as the introduction of the carbon pricing and the local economic level?

Some references are premature, please use more recent references.

Author Response

Reviewer 1

Removing the fossil fuel subsidies and imposing the carbon pricing have been discussed for more than a decade, but their potential on the reduction of carbon emission is still unclear. In this work, the simulation model, namely GSI-IF, was designed to assess the emission reduction potential resulting from removing fossil fuel subsidies and recycling part of the avoided subsidy and additional revenue from carbon pricing to renewable energy and energy efficiency. In my opinion, the work is suitable for publication in World. However, the following remarks are carefully addressed.

  • What is the difference between this work and the previous study? And what is the practical significance of this work?

Thank you for your question, this is an important clarification to make. We find that the vast majority of the other studies that assessed the impact of subsidy removal on emissions did not consider the possibility to reallocate part of the subsidy savings to investments in energy efficiency and renewable energy, to further reduce emissions. We also distinguish between the impact of energy efficiency (which reduces the energy cost for households by lowering consumption and emissions) versus the impact of renewable energy (which reduced future costs, not current ones, by lowering the impact of future carbon pricing). There are differentiated impacts, and these can offer insights to policymakers that are planning intervention options for the achievement of NDC targets.

Also, we believe that it is important to offer this type of assessment now, in a situation where energy prices have increased worldwide due to conflicts and supply chain challenges, while at the same time there is ambition to reduce emissions towards the NDC targets and even net zero targets by 2050. Subsidy removal is a challenging policy to implement, but there important gains that often go unnoticed, especially when market prices are high (which is when investments in energy efficiency, for instance, offer the most value).

I have added this text in the abstract:

In the current context (year 2022) with high energy prices, heavy stress on fiscal balances, and of the renewed ambition of most government to reduce emissions towards Net Zero in 2050, subsidy removal and carbon pricing hold promise in the toolbox of decarbonization options while im-proving fiscal sustainability.

I have added this text in the introduction to clarify:

Over the last three decades, fossil fuel subsidy reform has been recognised as an integral tool not only to reduce GHG emissions. , On the other hand, we find that the vast majority of studies that assessed the impact of subsidy removal on emissions did not consider the possibility to reallocate part of the subsidy savings to investments in energy efficiency and renewable energy, to further reduce emissions. Further, it is critical to distinguish between the impact of energy efficiency (which reduces the energy cost for households by lowering consumption and emissions) versus the impact of renewable energy (which reduces future costs, not current expenditure, by lowering the impact of future carbon pricing). There are differentiated impacts, and these can offer insights to policymakers that are planning intervention options for the achievement of NDC targets. but also to encourage energy efficiency, reduce the fiscal burden of governments, cut energy imports and energy-related costs, and curtail environmental damage [1]. This re-search builds, and extends on This can be seen in the expansive number of studies on the topic, varying in geographical scope, definition and calculation of subsidies, assumptions, sectors and fuels under consideration, reform period, type of subsidy and the model leveraged for analysis [1].

  • The modeling process and the structure of GSI-IF model should be specifically described.

The model is described in section 3. An annex could be provided with all the equations (a few, key equations are included in section 3). Or is there a specific area of the model that is currently not sufficiently described and documented in section 3 and related sub-sections?

  • What is the relationship between the removal of the fossil fuel subsidies as well as the introduction of the carbon pricing and the local economic level?

This is a very good question. We have added the following text in section 4.

A consideration has to be made in relation to energy affordability when removing fossil fuel subsidies and introducing carbon pricing. First, the scenarios of reallocation of subsidy savings only consider that 30% of this amount will be invested in energy efficiency and renewable energy. The remaining 70% could be used to support low income families, or the reduction of public deficit. Second, while the GSI-IF model does not estimate the impact of energy prices on energy spending, and hence on affordability, it estimates changes in energy consumption as a result of price changes. The change in energy consumption could be considered a proxy for energy affordability. Overall, we find that the introduction of subsidy removal and carbon taxes increases energy prices, but energy efficiency reduces consumption by an amount larger than the price increase. Over time, the affordability of energy would increase as a result (i.e. if energy consumption declines more than the increase in price, affordability will increase), but it is important to plan a phased removal of subsidies, and a simultaneous improvement in energy efficiency.

Some references are premature, please use more recent references.

We have reviewed the references used, and we believe that we have used the most appropriate ones. Most of the old references are related to the documentation of simulation models, for which we have used the original project report or first paper in the literature. Otherwise we have tried to cover a broad spectrum of studies, which can be found primarily in the period 2009-2010 (as a result of the high fossil fuel prices  until 2008 and sharp decline thereafter), 2015 (for issues with fiscal balance sustainability) and 2019-2021 (for renewed commitments and the preparation of NDCs).

Reviewer 2 Report

Interesting analysis for greenhouse gases produced by fossil fuels and its relation to pricing is presented in this manuscript. However few modifications are needed for this article to be accepted.

1. Please add the table of acronyms, that would simplify the text comprehension

2. The chemical formula for is  CO2  rather than CO2 please correct it throughout the text

3. Please separate tCO2 to t COor t of COsame with Gt

4. The GSI-IF model is a System Dynamics based simulation model developed to fore- 247 cast energy demand and related emissions for a number of countries. The model projects 248 national energy consumption by sector and source, from 1990 to 2040. GHG emissions are 249 estimated based on the Energy demand projections generated by the model and IPCC 250 GHG multipliers by fuel source [37].  Please specify that such approach accuracy might change, since widespread penetration of various technique to tackle greenhouse gas emissions. For example in the UK the companies allow consumers, to benefit from using the energy outside of peak hours. In addition there is penetration of dynamic, time of use tariffs (Octopus Energy Agile Tarif) and  generation following tariff for example FanClub from Octopus Energy. Moreover there are multiple system that could follow the greenhouse gases emissions predication by the grid in the area or other signals (Shukhobodskiy et al. 2021 https://doi.org/10.1016/j.jclepro.2021.128926 , Amini Toosi 2023https://doi.org/10.1016/j.apenergy.2023.120648 ). Last but not least, the penetration of Electric Vehicles also contribute to greenhouse gas emissions.  All these allow to deviate significantly from standard consumption profile as well as tackle the curtailment  of renewable energy sources. Please add the discussion with regards to the above.

5. Smart energy systems, have potential to significantly reduce greenhouse gas emission please add discussion for example that is significantly discussed in Kostopoulus, 2022 https://doi.org/10.3390/en15218214

Author Response

Reviewer 2

Interesting analysis for greenhouse gases produced by fossil fuels and its relation to pricing is presented in this manuscript. However few modifications are needed for this article to be accepted.

  1. Please add the table of acronyms, that would simplify the text comprehension

I would include the following acronyms, but I would not know where to add it in the paper, and if it is accepted. I would normally include it in reports, not in peer-reviewed papers. Apologies for my ignorance on this, I will seek guidance from the editor since I could not find other papers that include a list of acronyms for this journal.

  1. The chemical formula for is CO2  rather than CO2 please correct it throughout the text

Thank you, I have implemented this change throughout the paper.

  1. Please separate tCO2 to t CO2 or t of CO2 same with Gt

Thank you, I have implemented this change also, throughout the paper.

  1. The GSI-IF model is a System Dynamics based simulation model developed to forecast energy demand and related emissions for a number of countries. The model projects national energy consumption by sector and source, from 1990 to 2040. GHG emissions are estimated based on the Energy demand projections generated by the model and IPCC GHG multipliers by fuel source [37].

Please specify that such approach accuracy might change, since widespread penetration of various technique to tackle greenhouse gas emissions. For example in the UK the companies allow consumers, to benefit from using the energy outside of peak hours. In addition there is penetration of dynamic, time of use tariffs (Octopus Energy Agile Tarif) and  generation following tariff for example FanClub from Octopus Energy. Moreover there are multiple system that could follow the greenhouse gases emissions predication by the grid in the area or other signals (Shukhobodskiy et al. 2021 https://doi.org/10.1016/j.jclepro.2021.128926 , Amini Toosi 2023https://doi.org/10.1016/j.apenergy.2023.120648 ). Last but not least, the penetration of Electric Vehicles also contribute to greenhouse gas emissions.  All these allow to deviate significantly from standard consumption profile as well as tackle the curtailment  of renewable energy sources. Please add the discussion with regards to the above.

Thank you, this is a very relevant comment. The model forecasts energy consumption by sector and energy source, and it may not consider the emergence of new practices and technologies. Also, we only consider investments in energy efficiency and power generation from renewable energy. More options to could be considered.

I have added text as follows:

When interpreting these results it should also be considered that the price of low carbon technologies is rapidly declining and that their adoption is likely to accelerate in the coming years due international commitments for emission reduction. The adoption of such technologies may affect the business as usual scenario by reducing emissions, and hence potentially reduce the net decline in emissions that is achieved with subsidy re-moval and carbon pricing. Practically, if technologies are already becoming more eco-nomically viable, the price signal sent by subsidy removal and carbon pricing may be-come less effective in determining the uptake of new technologies and practices, but it would nevertheless support this process.

We decided not to mention specific references, because there are many options that could be considered, and pointing to one or two specific options would not be accurate.

  1. Smart energy systems, have potential to significantly reduce greenhouse gas emission please add discussion for example that is significantly discussed in Kostopoulus, 2022 https://doi.org/10.3390/en15218214

See response above, for the integration of the discussion on possible deviations in the BAU scenario. Thank you for all your comments and suggestions, it is much appreciated.

Reviewer 3 Report

Good empirical work. its potential suitability for publication in World (MDPI). However, some issues should be addressed before publication.

1. Title needs to be rewritten.

2. In the abstract part need to rewrite also mention the time of the study.

3. Introduction part is not strong, the author needs to make a strong introduction with the research gap, research question, the contribution of the study with evidence, and also discuss the using model. And last part of the introduction discusses the research content of this study.

4. Need to give equation number

5. Conclusion is written very simply way,  need to write a strong conclusion

Author Response

Reviewer 3

Good empirical work. its potential suitability for publication in World (MDPI). However, some issues should be addressed before publication.

  1. Title needs to be rewritten.

Thank you for your review and useful comments. I have exchanged with all co-authors and the preference is to keep the current title. If not acceptable, could you explain your concern? This would be helpful in determining how to improve it. We want to focus on the policy intervention (on the pricing side), and stress the synergy emerging from the reallocation of subsidy savings. Thank you.

  1. In the abstract part need to rewrite also mention the time of the study.

Thank you, it is important to frame the timing of the study. We have edited the text of the abstract as follows:

In the current context (year 2022) with high energy prices, heavy stress on fiscal balances, and of the renewed ambition of most government to reduce emissions towards Net Zero in 2050, subsidy removal and carbon pricing hold promise in the toolbox of decarbonization options while im-proving fiscal sustainability.

  1. Introduction part is not strong, the author needs to make a strong introduction with the research gap, research question, the contribution of the study with evidence, and also discuss the using model. And last part of the introduction discusses the research content of this study.

In order to strengthen the abstract and introduction, I have added this text in the abstract:

In the current context (year 2022) with high energy prices, heavy stress on fiscal balances, and of the renewed ambition of most government to reduce emissions towards Net Zero in 2050, subsidy removal and carbon pricing hold promise in the toolbox of decarbonization options while im-proving fiscal sustainability.

I have added this text in the introduction to clarify:

Over the last three decades, fossil fuel subsidy reform has been recognised as an integral tool not only to reduce GHG emissions. , On the other hand, we find that the vast majority of studies that assessed the impact of subsidy removal on emissions did not consider the possibility to reallocate part of the subsidy savings to investments in energy efficiency and renewable energy, to further reduce emissions. Further, it is critical to distinguish between the impact of energy efficiency (which reduces the energy cost for households by lowering consumption and emissions) versus the impact of renewable energy (which reduces future costs, not current expenditure, by lowering the impact of future carbon pricing). There are differentiated impacts, and these can offer insights to policymakers that are planning intervention options for the achievement of NDC targets. but also to encourage energy efficiency, reduce the fiscal burden of governments, cut energy imports and energy-related costs, and curtail environmental damage [1]. This re-search builds, and extends on This can be seen in the expansive number of studies on the topic, varying in geographical scope, definition and calculation of subsidies, assumptions, sectors and fuels under consideration, reform period, type of subsidy and the model leveraged for analysis [1].

  1. Need to give equation number

We have added the numbering of equations, thank you.

  1. Conclusion is written very simply way, need to write a strong conclusion

We have expanded the text in the conclusions, mentioning the following:

We find that the potential reduction in emissions emerging from the reallocation of 30% of subsidy savings to energy efficiency and renewable energy is potentially higher that the reduction caused by subsidy removal and carbon pricing, especially in countries characterized by high energy demand. This indicates a strong synergy with other climate mitigation options, and an important cost-reduction potential, in the face of possibly higher energy prices.

Round 2

Reviewer 2 Report

Thank you for the edits, it has significantly improved the article however few major thing is still needed to be addressed.  “ We decided not to mention specific references, because there are many options that could be considered, and pointing to one or two specific options would not be accurate” if there are too many options, I would suggest to mention comprehensive reviews, unreferenced statements are not acceptable in scientific publications. Please  add some references, if there are too many references it is better to list all. 

Author Response

Thank you, comment well received. We have identified and added two references, one from Nicholas Stern (academic paper, policy oriented) and a more technical paper on modeling and implications of decarbonizations trends (IMF report, authors my a team of modelers). 

These are references 44 and 45, added in section 5, in the paragraph that introduces the importance of considering the potential uptake of new technology. 

Many thanks again, and best regards.

Round 3

Reviewer 2 Report

I would like to thank authors for modifications and recommend accept article as it is